# The $5\alpha$ condensate state in $^{20}$Ne

Bo Zhou [1,2] ✉, Yasuro Funaki [3] ✉, Hisashi Horiuchi[4], Yu-Gang Ma[1,2], Gerd Röpke [5], Peter Schuck[6], Akihiro Tohsaki[4] & Taiichi Yamada[3]

The formed $^4$He ($\alpha$) clusters consisting of two neutrons and two protons can be a building block in light nuclear systems. Intriguingly, these alpha clusters could potentially form alpha condensate states within the nuclear system. The Hoyle state at 7.65 MeV in $^{12}$C, which plays an essential role in stellar nucleosynthesis, is now considered to be a phase transition, namely the $3\alpha$ Bose-Einstein condensate. Confirming the existence of Hoyle-analog states in $N\alpha$ nuclei ($N > 3$) remains a major challenge. Here we show microscopic five-body calculations for the $^{20}$Ne nucleus. We find that one excited $0^+$ state has a distinct gas-like characteristic and represents the condensate state. Identifying the $5\alpha$ condensate state is an important step in establishing the concept of $\alpha$ condensation in nuclear fermion systems.

The observed atomic Bose-Einstein condensation (BEC)[1] in 1995 opened a significant era for the study of BEC in various Bose systems, from dilute atomic gases to quasi-particles in solids. Since then there have been many speculations[2,3] whether such an exotic many-body quantum phenomenon can also occur in atomic nuclei. As early as the 1930s, Gamow et al. proposed[4] that $\alpha$-conjugate nuclei such as $^{12}$C, $^{16}$O, and $^{20}$Ne, are composed of $\alpha$ particles. This idea is obviously oversimplified, but since the 1960s many studies[5–9] have shown that the concept of $\alpha$ clustering is essential for understanding the structure of light nuclei. Based on the Ikeda diagram[10], the evolved $N\alpha$ clustering structure could occur around the thresholds for the $N\alpha$ breakup of $\alpha$-conjugate nuclei. Recent studies[11] have also shown that above the thresholds, the $3\alpha$ and $4\alpha$ clusters exhibit a variety of exotic phenomena such as gas-like and linear-chain clustering. On the other hand, based on studies for nuclear matter[12], it has been found in recent years that the $\alpha$ clusters can jointly occupy the lowest (0S) orbit when the density is below one-fifth of the saturation density. A natural question is whether we can find the $\alpha$ condensate states in finite nuclei. The existence of the $N\alpha$ condensate extends our knowledge of the fundamental nuclear interaction and nuclear structure. Meanwhile, it could increase the symmetry energy of nuclear matter and finally have a great impact on the equation of state of nuclear matter, which is closely related to astrophysical questions[13–16]. In this context, knowledge of the density dependence of the symmetry energy is crucial for understanding the collapse of supernovae and the properties of neutron stars resulting from supernova collapses[17,18].

In 2001, the existence of $\alpha$ condensates in finite nuclei was proposed by means of the THSR (Tohsaki-Horiuchi-Schuck-Röpke) wave function[19], which is analogous to the BCS (Bardeen-Cooper-Schrieffer) wave function replacing the Cooper pairs by $\alpha$ particles (quartets). The Hoyle state of $^{12}$C, playing a central role for nucleosynthesis, is also known for its well-developed $3\alpha$ clustering structure and it has become a touchstone for nuclear structure[20]. One striking fact[21,22] is that the single THSR wave function of the Hoyle state is found to be almost equivalent to the full solution of microscopic $3\alpha$ problem, i.e. the wave functions of resonating group method (RGM)/generator coordinate method (GCM). The Hoyle state can be the $3\alpha$ Bose-Einstein condensate state in the nuclear system. The $3\alpha$ clusters could move almost independently and the occupation of the (0S) center-of-mass wave function of the $\alpha$ particle is over 70%[23,24]. The remaining fewer 30% non-bosonic product states are from the weak antisymmetrization[25]. It is believed that the occurrence of this peculiar state is not just a lucky coincidence[20,26]. This triggered the intense search for the $N\alpha$ condensate in atomic nuclei, both experimentally[27–29] and theoretically[30]. Theoretical studies[31–33] show that the $0_6^+$ state of $^{16}$O is a strong candidate for $4\alpha$ condensate and great experimental efforts are being made using sequential decay measurement to confirm it. In ref. 27, it was shown that the $N\alpha$ condensate state has an enhanced preference for the emission of gas-like or $(N-1)\alpha$ condensate states, which would

[1]Key Laboratory of Nuclear Physics and Ion-Beam Application (MoE), Institute of Modern Physics, Fudan University, 200433 Shanghai, China. [2]Shanghai Research Center for Theoretical Nuclear Physics, NSFC and Fudan University, 200438 Shanghai, China. [3]College of Science and Engineering, Kanto Gakuin University, Yokohama 236-8501, Japan. [4]Research Center for Nuclear Physics (RCNP), Osaka University, Osaka 567-0047, Japan. [5]Institut für Physik, Universität Rostock, D-18051 Rostock, Germany. [6]Institut de Physique Nucléaire, Université Paris-Sud, IN2P3-CNRS, UMR 8608, F-91406 Orsay, France. ✉e-mail: zhou_bo@fudan.edu.cn; yasuro@kanto-gakuin.ac.jp

be an experimental signature for the existence of the condensate. However, the predicted $0_6^+$ condensate state of $^{16}$O is less than 1 MeV above the $4\alpha$ threshold, i.e., this state is close to the $^{12}$C $(0_2^+)$ + $\alpha$ threshold. In this case, the $\alpha$ particle decaying into the channel $^{16}$O $(0_6^+) \rightarrow {}^{12}$C $(0_2^+)$ + $\alpha$ almost cannot be observed, due to the difficulty of penetrating through great Coulomb barrier. The calculated partial $\alpha$ decay width is only the order of $10^{-10}$ MeV[25].

Fortunately, it is shown that the energy of possible $N\alpha$ condensate does not always remain close to the $N\alpha$ threshold and in fact gradually increases with the $\alpha$-number $N$, which is due to the competition between the attractive nuclear potential and the repulsive Coulomb potential[34]. In comparison with $3\alpha$ and $4\alpha$ condensate states, the $5\alpha$ condensate state, if such a state exists, would appear somewhat higher, e.g., a few MeV above the $5\alpha$ threshold. The larger decay energy could be an important prerequisite to observe the decay of the $5\alpha$ condensate state. Recently, the experimental group lead by Kawabata[35] at the Osaka University performed the experiment of inelastic $^{20}$Ne $(\alpha, \alpha')$ reaction $(E_\alpha = 389$ MeV). They observed that three states at $E_x = 23.6$, 21.8, and 21.2 MeV in $^{20}$Ne are strongly coupled to the $0_6^+$ state in $^{16}$O. This provides an important clue to the $5\alpha$ condensate state of $^{20}$Ne. Meanwhile, Swartz et al.[36] performed reaction experiments $^{22}$Ne$(p, t)^{20}$Ne, and the excited states up to $E_x = 25$ MeV of $^{20}$Ne were studied at the iThemba LABS. They found that the state at $E_x = 22.5$ MeV cannot be interpreted by the shell-model calculations and could be the $5\alpha$ cluster state.

In this work, we perform the microscopic five-body calculations for studying the $5\alpha$ clustering structure in $^{20}$Ne. It is found that a $0^+$ state, which is around 3 MeV above the $5\alpha$ threshold, has a very large amplitude of the $^{16}$O $(0_6^+)$ + $\alpha$ structure, which shows a clear characteristic of $5\alpha$ condensate state. It is further shown that the observed $\alpha$ decay provides a remarkable link between the $5\alpha$ condensate and $4\alpha$ condensate states. This $5\alpha$ condensate state we found could correspond to one observed state in the recent experiment.

## Results

### Two obtained $0^+$ states above the $5\alpha$ threshold

The $^{20}$Ne nucleus is well known for its rich clustering structure and has been studied for more than half a century[37–39]. With the increase of excitation energy, the 20 nucleons are more favorable to be rearranged from the liquid-like ground-state structure to form different clustering structure, such as the $^{16}$O + $\alpha$ and $^{12}$C + $^8$Be clustering, and could evolve to the $5\alpha$ gas-like structure around the $5\alpha$ threshold $(E_x = 19.2$ MeV) as shown in Fig. 1. Based on the threshold rule, it is at least energetically allowed that this kind of $5\alpha$ clustering structure appears.

However, for this kind of weakly-bound $5\alpha$ system, we have to deal with the five-body problem, which meets much more difficulties than with the three-body and four-body problems and is completely beyond the traditional cluster models. The proposed THSR wave function is particularly suitable for the description of the gas-like states and plays a central role in studying the $3\alpha$ and $4\alpha$ condensate states[30]. In this work, we construct the THSR-type wave function for investigating the $5\alpha$ problem. Details of the models are shown in the method part.

We performed full-microscopic calculations for the $5\alpha$ structure and obtained 19 states of $J^\pi = 0^+$ in $^{20}$Ne. Among the obtained 19 states, the most significant ones are those above the $5\alpha$ threshold, as they are potential candidates for the $5\alpha$ condensate state. Our current calculations yield five $0^+$ states $(0_{15-19}^+$ states) above the threshold. However, two of these five, $0_{17}^+$ and $0_{19}^+$ eigenstates, are only reasonably considered as candidates for $5\alpha$ cluster states. The remaining three $0^+$ states are considered to be unphysical due to contamination from continuum states, based on the radius constraint method and analysis of their reduced width amplitudes in different channels. For further details, please refer to the method section. In the next discussions, we only need to focus on the $0_{17}^+$ and $0_{19}^+$ states, which are denoted as $0_I^+$

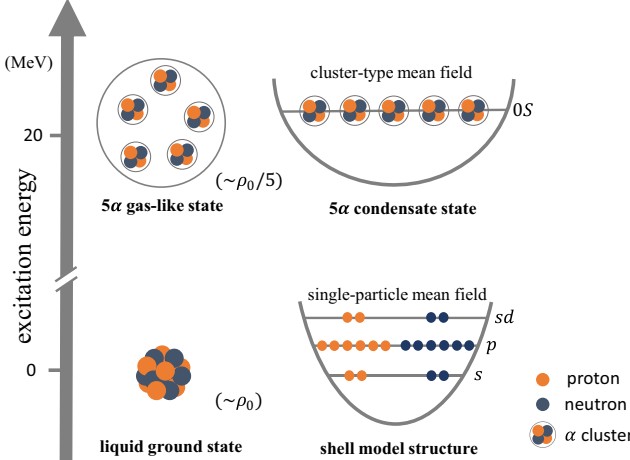

**Fig. 1 | Diagrammatic representation for the shell-model-like ground state and the possible $5\alpha$ condensate state in $^{20}$Ne.** The ground state of $^{20}$Ne, with the saturated density $\rho_0$, has a liquid compact structure and it can be described by the standard shell-model picture, in which the nucleons are assumed to move in a single-particle mean field and occupy different orbits. With the increase of excitation energy, the liquid-like ground state can be evolved to various clustering structures. Around the $5\alpha$ threshold $(E_x = 19.2$ MeV) and the low density e.g., $\rho_0/5$, the $5\alpha$ clustering structure is expected to form BEC condensate state, in which the $5\alpha$ clusters mainly move with (0S) orbit in a cluster-type mean field.

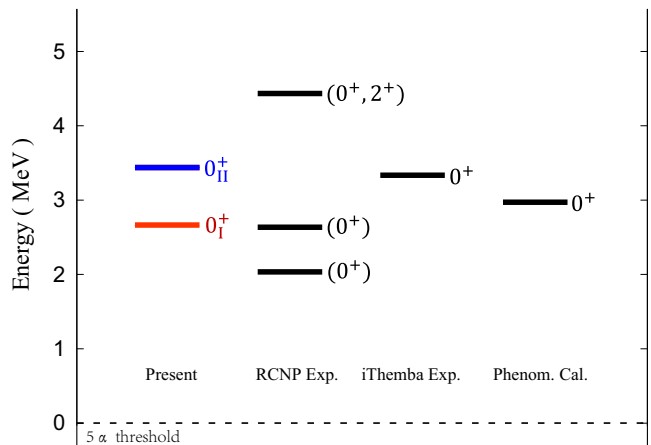

**Fig. 2 | Comparison of theoretical predictions with experimental results for the $5\alpha$ cluster states above the threshold.** The present obtained results are the $0_I^+$ state (red color) and $0_{II}^+$ state (blue color), which are 2.7 MeV and 3.4 MeV above the threshold, respectively. Experimental results are based on the RCNP (Research Center for Nuclear Physics) experiment[35] (Spins and parities ($J^\pi$) of states are not determined. Possible $J^\pi$ values are shown in bracket) and the iThemba LABS (Laboratory for Accelerator Based Sciences) experiment[36]. The phenomenological calculation[34] is also shown for comparison.

and $0_{II}^+$ states as shown in Fig. 2. First, both $0^+$ states are obtained at about 3 MeV above the $5\alpha$ threshold. The energies of these states are qualitatively in agreement with those observed in experiments. Indeed, the proper excitation energy (relative to $5\alpha$ threshold) is a prerequisite for the condensate state. Using the phenomenological calculations[34] of the Gross-Pitaevskii equation, it is shown that the total energy of the $N\alpha$ gas-like state gradually increases and the possible $5\alpha$ gas-like state could appear at about 3 MeV, as shown in Fig. 2. Note that the Hoyle state and the $0_6^+$ state of $^{16}$O appear at less than 1 MeV above their corresponding thresholds.

In the recent RCNP experiment[35], it is found from the observed branching ratio that three states as shown in Fig. 2 are strongly coupled to the candidate for the $4\alpha$ condensate state, suggesting that these

three states may all have the dominant $^{16}$O ($0_6^+$) + $\alpha$ configuration. Nevertheless, the spins and parities of these observed states have not been assigned. The state around 4.5 MeV could even be the excited $2^+$ state as explained in ref. 35. In the iThemba LABS experiment[36], the newly found $0^+$ state around 3.5 MeV cannot be interpreted by the shell model and it may be the $5\alpha$ cluster state. Unfortunately, the decay and structural information for this state is still unknown. As a whole, the current experiments provide important support for the $5\alpha$ condensate state, while some key information of the $5\alpha$ structure is still missing. The analysis of the $0_I^+$ state and $0_{II}^+$ state obtained in this energy range (~3 MeV) helps to solve the problem of the existence of the $5\alpha$ condensate state.

## The 5α condensate state

In experiments, a dominant decay channel to $^{16}$O ($0_6^+$) + $\alpha$ has been observed which is a strong support in identifying the $5\alpha$ condensate. Similarly, the calculated spectroscopic $S^2$ factors can be the direct way to analyze the $^{16}$O ($0_6^+$) + $\alpha$ structure. As we know, the $4\alpha$ condensate state of $^{16}$O has been studied for many years, and $0_6^+$ state is now

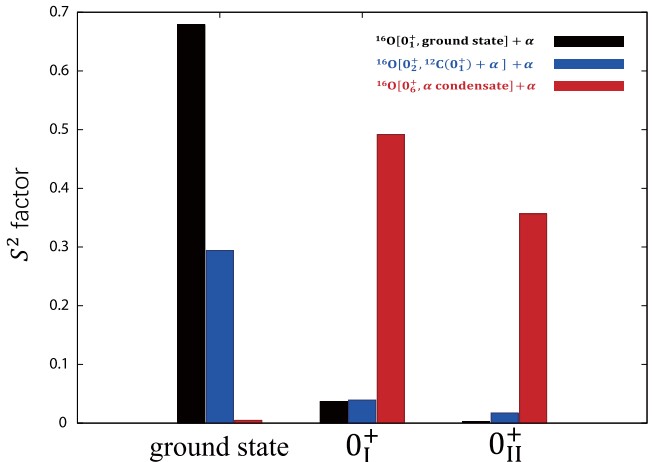

**Fig. 3 | Spectroscopic $S^2$ factor of $^{20}$Ne.** Based on the obtained wave functions of the ground state $0_{g.s.}^+$, $0_I^+$, and $0_{II}^+$ states of $^{20}$Ne, the spectroscopic $S^2$ factor for three channels of $^{16}$O + $\alpha$ are calculated. Three channels each are $^{16}$O ($0_{g.s.}^+$) + $\alpha$ (black), $^{16}$O ($0_2^+$) + $\alpha$ (blue), and $^{16}$O ($0_6^+$) + $\alpha$ (red).

considered as the $4\alpha$ condensate state[31,40]. Figure 3 shows calculated spectroscopic $S^2$ factors for $^{16}$O (ground state) + $\alpha$, $^{16}$O ($0_2^+$) + $\alpha$, and $^{16}$O ($0_6^+$) + $\alpha$ channels in the ground state and two $0^+$ states of $^{20}$Ne. As expected, the shell-model-like ground state of $^{20}$Ne has a large component of the compact $^{16}$O (ground state) + $\alpha$ configuration and some non-negligible $^{16}$O ($0_2^+$) + $\alpha$ components. For the $0_I^+$ and $0_{II}^+$ states, it is interesting to see that the $^{16}$O ($0_6^+$) + $\alpha$ configurations dominate in these two states. At the same time, these two states have very small fractions of the $^{16}$O (ground state) + $\alpha$ and $^{16}$O ($0_2^+$) + $\alpha$ components. This suggests that the $0_I^+$ and $0_{II}^+$ states both have a very large overlap with the $^{16}$O ($0_6^+$) + $\alpha$ configuration. We should note that the dominance of $^{16}$O ($0_6^+$) + $\alpha$ structure means that the $\alpha$ cluster can move around the $^{16}$O ($0_6^+$) core, and only if the outer $\alpha$ cluster mainly sits in the (0S) orbit, this configuration corresponds to the $5\alpha$ condensate structure. Thus, to identify the $5\alpha$ condensate state, the character of the relative wave function of the $4\alpha$ core and the outer $\alpha$ cluster should be clarified in more detail.

Figure 4 shows the calculated reduced width amplitudes (RWA) of the $0_I^+$ state and $0_{II}^+$ state in the channel of $^{16}$O ($0_6^+$) + $\alpha$, which can show us the behavior of the relative wave function of $^{16}$O ($0_6^+$) and $\alpha$ in $^{20}$Ne. It can be clearly seen that the two $0^+$ states have obviously larger amplitudes compared to other channel components (See the Supplementary Fig. 4). In particular, the $0_I^+$ state has a rather large amplitude around 6 fm and a long tail extending to 20 fm. The feature of the Gaussian-like RWA obtained here is quite similar to those of $3\alpha$ and $4\alpha$ condensate states. This type of RWA behavior with zero nodes and large amplitude is an important feature of the $\alpha$ condensate originating from the (0S) motion between clusters. On the other hand, the $0_{II}^+$ state has a relatively small amplitude in the inner region and a peak around 20 fm in the outer region with a strongly extended tail. Most importantly, it has one node in the RWA, suggesting that this state could be the excited state of the $0_I^+$ state. Therefore, the $0_I^+$ state ($E_x \approx 22$ MeV) we obtained can be the strong candidate for $5\alpha$ condensate state.

## Another approach to pin down the 5α condensate state

Besides the predominant $4\alpha$ (condensate state) + $\alpha$ structure, the $5\alpha$ condensate state itself has a peculiar $5\alpha$ gas-like structure, in which the $\alpha$ particles can move relatively free in a cluster-type mean field[41] as shown in Fig. 1. This picture suggests that the obtained wave functions of condensate state should have a larger overlap with the single one-$\beta$ THSR wave functions with larger value of size variable $\beta$. This is an important and simple idea to identify the $\alpha$ condensate state

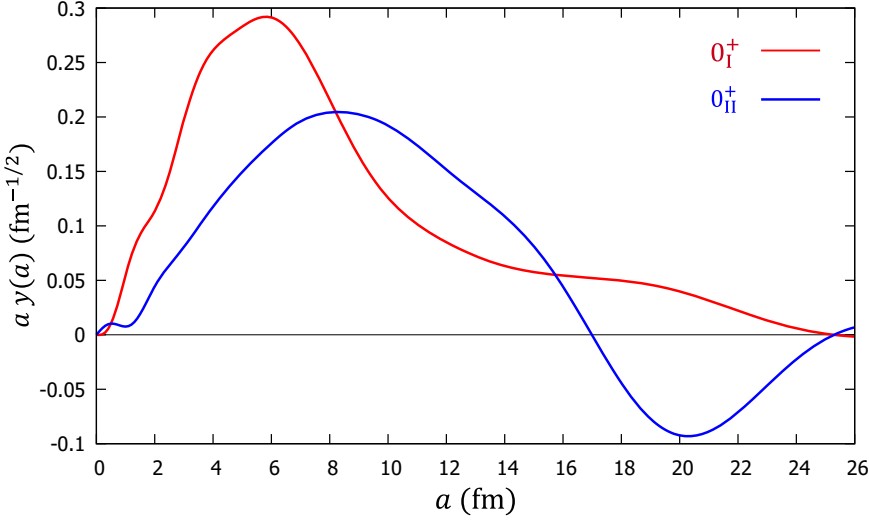

**Fig. 4 | The calculated reduced width amplitudes of the $0_I^+$ state and $0_{II}^+$ state in $^{20}$Ne in the channel of $^{16}$O ($0_6^+$) + $\alpha$.** The horizontal axis $a$ can be considered to represent the distance between the $\alpha$ and $^{16}$O ($0_6^+$) clusters. The vertical coordinates represent $ay(a)$ and $y(a)$ is the reduced width amplitude defined in Eq. (8). The $ay(a)$ curves of $0_I^+$ state and $0_{II}^+$ state are shown in red color and blue color, respectively.

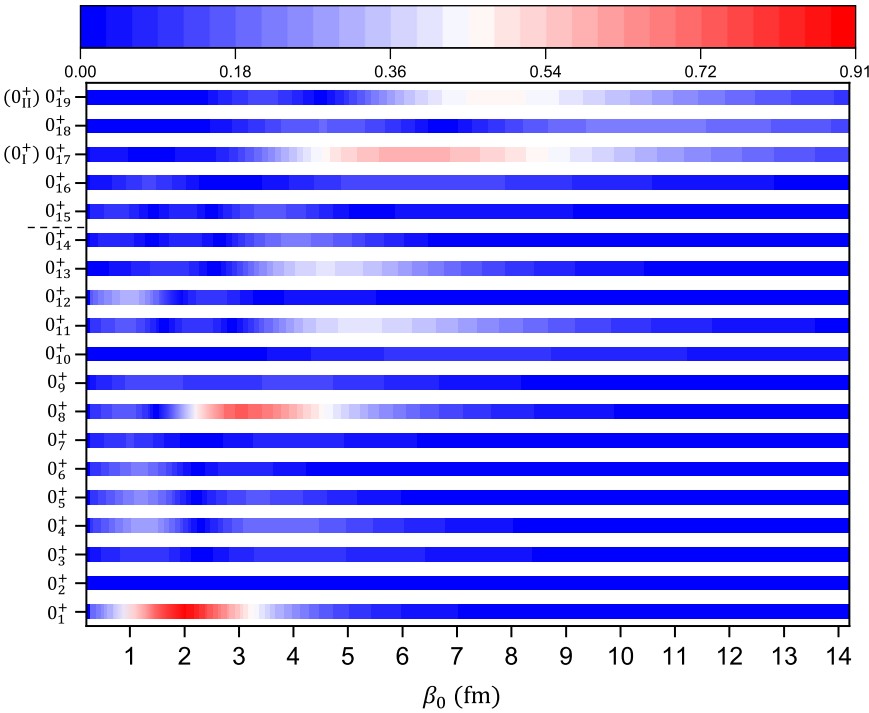

**Fig. 5 | Split heatmap for contour plots of overlap.** The overlap $|\langle\Phi^{0^+}(\beta_0)|\Psi_{gcm}^{0_\lambda^+}\rangle|$ between the two-$\beta$ GCM THSR wave functions and the single one-$\beta$ THSR wave function for $0_\lambda^+$ states of $^{20}$Ne are calculated. The $0_\lambda^+$ states are labeled in the vertical axis. The $\beta_0$ represents the size variable in the THSR wave function $\Phi^{0^+}(\beta_0)$. For the obtained $0^+$ states, the $0_I^+$ and $0_{II}^+$ states are corresponding the $0_{17}^+$ and $0_{19}^+$ states, respectively. Between the $0_{14}^+$ and $0_{15}^+$ states in the vertical axis, the short dashed line represents the $5\alpha$ threshold.

theoretically. Without restriction of generality, we can perform an analysis of all the $0^+$ states obtained. Figure 5 shows the contour plot for the obtained 19 eigenstates with $J^\pi = 0^+$ by calculating their overlap $|\langle\Phi^{0^+}(\beta_0)|\Psi_{gcm}^{0_\lambda^+}\rangle|$ ($\lambda = 1, \cdots, 19$). $\Phi^{0^+}(\beta_0)$ is the normalized THSR wave function with $\beta_1 = \beta_2 = \beta_0$ in Eq. (1). It is clear that, above the $5\alpha$ threshold, the $0_I^+$ ($0_{17}^+$) state ($E_x \approx 22$ MeV) is distinguished by its larger value of overlap. At $\beta_0 \approx 6$ fm, the overlap value is about 0.6, which is much higher than those for the neighboring states. It should be noted that, if we consider orthogonality, we construct one single wave function that is orthogonal to the wave functions of states below the $0_I^+$ state as $\Phi_\perp^{0^+}(\beta_0) = N_0(1 - \sum_{i=1}^{16}|\Psi_{gcm}^{0_i^+}\rangle\langle\Psi_{gcm}^{0_i^+}|)\Phi^{0^+}(\beta_0)$, where $N_0$ is a normalization factor. Then, the overlap $|\langle\Phi_\perp^{0^+}(\beta_0)|\Psi_{gcm}^{0_{17}^+}\rangle|$ is as high as 0.8. As we have shown in Figs. 3 and 4, the $0_{II}^+$ state has some similarities with the $5\alpha$ condensate state and it even has a longer tail. This point can be reflected in the contour plot. Below the $5\alpha$ threshold, with the increase of $\beta_0$, we can see the ground state ($\beta_0 \approx 2$ fm) and the $0_8^+$ state ($\beta_0 \approx 3$ fm) have larger values of overlap. Additionally, the $0_{II}^+$ and $0_{13}^+$ states ($\beta_0 \approx 5$ fm) also exhibit some non-negligible components with the single THSR wave functions. These can be regarded as the intermediate states that evolve into the ultimate gas-like condensate state. In fact, most $0^+$ states have a quite small overlap with the one-$\beta$ THSR wave function, which is due to the non-$5\alpha$ clustering structure. It is therefore surprising that this simple one-$\beta$ container picture even provides a qualitative interpretation and identification for the structure of $5\alpha$ condensate state among these $0^+$ states. This analysis of the overlap gives more direct evidence that the $0_I^+$ state around 3 MeV above the threshold is exactly the $5\alpha$ condensate state we are looking for.

## The $\alpha$ decay from condensate states

The clustering structure of $^{16}$O and the predicted $^{16}$O ($0_6^+$) condensate state have been studied for many years. However, the observable

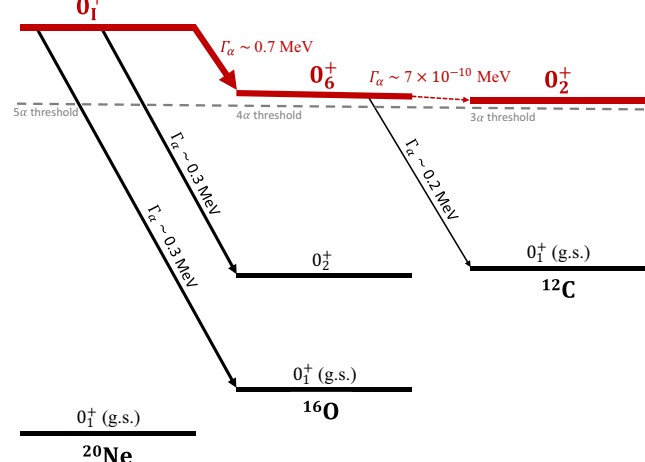

**Fig. 6 | The $\alpha$-decay scheme of condensate states.** The energy spectrum for the ground states and some excited states of $^{12}$C, $^{16}$O, and $^{20}$Ne are shown, which are relative to their corresponding thresholds. Red lines represent the candidates for $\alpha$ condensate states. The partial $\alpha$ decay widths for $4\alpha$ and $5\alpha$ condensate states are calculated and shown.

quantities for this $^{16}$O ($0_6^+$) state are rare and, in particular, the decay connection with the Hoyle state cannot be established due to the extremely small $\alpha$ partial decay width as shown in Fig. 6. Fortunately, the calculated partial $\alpha$ decay width of the predicted $5\alpha$ condensate state into the $4\alpha$ condensate state is as high as 0.7 MeV due to the increased excitation energy. This is a billion times stronger than the decay width of the $^{16}$O ($0_6^+$) state decaying into the Hoyle state. Thus, this dominated decay channel can be measured directly in experiment. The decay widths to the $0_2^+$ state and the ground state of $^{16}$O are also comparable and large enough. In fact, the RCNP experiment has shown

some states with this decay character. This information may help experiments to determine the $0_I^+$ state ($E_x \approx 22$ MeV) we predict, which clearly has condensate character based on the present theoretical analysis.

### Monopole transitions

The monopole transition[42] is another important quantity for identifying the cluster states in experiments. The calculated value of $M(E0)$ between the predicted condensate state $0_I^+$ and the ground state is -1 fm$^2$, which is similar to the monopole transition for the $0_6^+$ state of $^{16}$O. More interestingly, the strength of the monopole transition between the $0_I^+$ and $0_{II}^+$ states is as high as 24 fm$^2$ as shown in Table 1. This enhanced monopole transition strength suggests that this $0_{II}^+$ state could be the corresponding breathing-like state of the $5\alpha$ condensate state, similar to how the $0_3^+$ state of $^{12}$C can be the breathing-like state of Hoyle state[43]. To deal with more complicated highly excited states above the $5\alpha$ threshold, deformation and further $\alpha$ correlations should be considered in future work.

## Discussion

We have performed microscopic calculations for the five-body cluster system in $^{20}$Ne. Two $0^+$ eigenstates above the $5\alpha$ threshold are obtained. Based on the analysis of $^{16}$O ($0_6^+$) + $\alpha$ and $5\alpha$ cluster constituents, it was found that one state around 22 MeV is a strong candidate for the $5\alpha$ condensate state. This condensate state could be the $5\alpha$ state observed in recent experiments. It is strongly recommended that the monopole transition and $\alpha$ decay widths for this state should be measured in future experiments. The decay connection between the $5\alpha$ condensate state and the $4\alpha$ condensate state is demonstrated. This suggests that the Hoyle state characterized as the $3\alpha$ condensate is not a random event in $^{12}$C and analogous condensate states could be found in heavier nuclei under similar conditions.

## Methods

### The THSR-type wave function

Two decades after the introduction of the original THSR wave function[19], the techniques for solving multi-cluster systems have been greatly improved[22]. The subsequently proposed container picture[39,41,44] provides an approach to the description of the $\alpha$ condensate. This finally allowed us to treat the five-body cluster problem in our microscopic model. In order to treat the complex five-cluster system including the $4\alpha+\alpha$ configuration simultaneously, the 20-nucleon cluster wave function is constructed as follows,

$$\Psi(\beta_1,\beta_2) = \int d^3R_1 d^3R_2 d^3R_3 d^3R_4 d^3R_5$$
$$\times \exp\left[-\frac{1/2S_1^2 + 2/3S_2^2 + 3/4S_3^2}{\beta_1^2} - \frac{4/5S_4^2}{\beta_2^2}\right]\Phi^B(R_1,R_2,R_3,R_4,R_5) \quad (1)$$

$$= n_0 \mathscr{A}\left\{\exp\left[-\frac{2\xi_1^2 + 8/3\xi_2^2 + 3\xi_3^2}{2(b^2+2\beta_1^2)}\right]\exp\left[-\frac{16/5\xi_4^2}{2(b^2+2\beta_2^2)}\right]\prod_{i=1}^5 \varphi_i^{\text{int}}(\alpha)\right\}, \quad (2)$$

where the conventional Brink cluster wave function $\Phi^B$,

$$\Phi^B(R_1,R_2,R_3,R_4,R_5) = \frac{1}{\sqrt{20!}}\mathscr{A}[\phi_1(R_1)\dots\phi_5(R_2)\dots\phi_{20}(R_5)] \quad (3)$$

$$\propto \phi_g \mathscr{A}\left\{\exp\left[-\frac{2(\xi_1-S_1)^2 + 8/3(\xi_2-S_2)^2 + 3(\xi_3-S_3)^2}{2b^2}\right]\right.$$
$$\left.\times \exp\left[-\frac{16/5(\xi_4-S_4)^2}{2b^2}\right]\prod_{i=1}^5 \varphi_i^{\text{int}}(\alpha)\right\}, \quad (4)$$

with the single-nucleon wave function,

$$\phi_i(R_k) = \left(\frac{1}{\pi b^2}\right)^{\frac{3}{4}}\exp\left[-\frac{1}{2b^2}(r_i - R_k)^2\right]\chi_i\tau_i. \quad (5)$$

Here, $\phi_i(R_k)$ is the single-nucleon wave function characterized by the Gaussian center parameter $\{R_k\}$ and harmonic oscillator size parameter $b$. $\chi_i$ and $\tau_i$ are the spin and isospin parts, respectively. $\phi_g$ is the center-of-mass wave function. $\varphi_i^{\text{int}}(\alpha)$ is the intrinsic wave function of the $\alpha$ cluster. $n_0$ is a trivial factor from multi-dimensional integration. $\Phi^B(R_1, \cdots, R_5)$ is the conventional Brink cluster wave function[45] for $^{20}$Ne. To remove the center-of-mass effect, the generator coordinates $\{R_k\}$ can be easily transformed into $\{S_k\}$ with $S_k = R_{k+1} - 1/k\sum_{i=1}^k R_i$ ($k=1-4$), see the schematic diagram in Fig. 7. The introduced Jacobi coordinates $\{S_k\}$ in Eq. (1) are also very helpful to construct the container model mathematically. $\xi_i$ represent the Jacobi coordinates to describe the dynamics of $5\alpha$ clusters $\{X_i^\alpha\}$, i.e., $\xi_k = X_{k+1}^\alpha - 1/k\sum_{i=1}^k X_i^\alpha$ ($k=1-4$). As we know, the conventional Brink cluster model is difficult to apply to describe the $5\alpha$ gas-like states due to the large number of degree of freedom from $5\alpha$ clusters. As shown in Eq. (1) and Fig. 7, after analytical integration of five generator coordinates $\{R_k\}$, only the $\beta_1$ and $\beta_2$ generator coordinates remain as the introduced key size parameters constraining the motions of $4\alpha$ and $\alpha$ $-4\alpha$, respectively. Most importantly, this container picture characterizing the nonlocalized clustering is particularly suitable to describe the gas-like cluster states. Taking the spherical $\beta_1$ and $\beta_2$, our constructed positive-parity THSR-type wave function can be applied to describe the ground states and excited $0^+$ states in GCM (generator coordinate method) calculations without performing any heavy angular-momentum projections. Moreover, in the limit of $\beta_1 = \beta_2 \to 0$, the constructed THSR wave function coincides with the SU(3) shell model wave function for the description of the ground state. While on the other limit $\beta_1 = \beta_2 \to \infty$, the wave function becomes the simple product of five wave functions of $\alpha$ clusters, in which $5\alpha$ clusters are completely free to move around and there is no antisymmetric effect and interaction among clusters. This wave function has been specially designed,

### Table 1 | Monopole transition matrix elements of $^{20}$Ne

| transition states $^{20}$Ne | $0_I^+ \to 0_{g.s.}^+$ | $0_{II}^+ \to 0_{g.s.}^+$ | $0_{II}^+ \to 0_I^+$ |
|---|---|---|---|
| $M(E0)$ (fm$^2$) | 1.0 | 1.3 | 24 |

The monopole transition matrix elements $M(E0)$ between the $0_I^+$ state and the ground state, the $0_{II}^+$ state and the ground state, and the $0_I^+$ state and the $0_{II}^+$ state, are calculated.

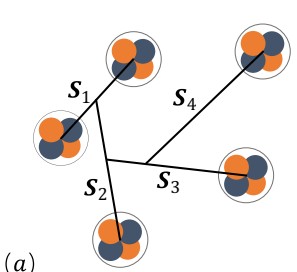
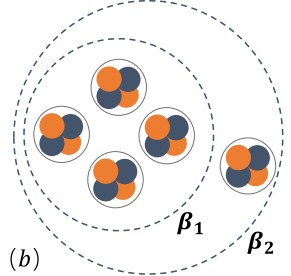

**Fig. 7 | Schematic illustrations of two distinct microscopic cluster models.** **a** The conventional cluster model of $\Phi^B$, in which the inter-cluster variables $\{S_i\}$ are the Jacobi coordinates of $\{R_i\}$. **b** Container picture for $4\alpha + \alpha$ cluster structure of $^{20}$Ne. The $\beta_1$ is the size variable for the description of $4\alpha$ and $\beta_2$ for the description of the relative motion between $4\alpha$ and $\alpha$ clusters.

($a$) Conventional cluster model          ($b$) Container model

and it is almost the unique microscopic way at present to search for $5\alpha$ condensate states from the point of view of the physical picture and calculations.

## Hamiltonian

To overcome the long-standing problem of binding energies[46] for $^{12}$C and $^{16}$O from two-body forces in nuclear cluster physics, we are taking the three-body effective interaction[47] that can reproduce the binding energies of $^{12}$C, $^{16}$O, $^{20}$Ne, and the experimental $\alpha$-$\alpha$ scattering phase shift. The present Hamiltonian contains no adjustable parameter,

$$H = \sum_{i=1}^{20} T_i - T_G + \sum_{i<j}^{20} V_{ij}^C + \sum_{i<j}^{20} V_{ij}^{(2)} + \sum_{i<j<k}^{20} V_{ijk}^{(3)}. \tag{6}$$

Here $T_i$ is the $i_{th}$ nucleon kinetic energy operator and $T_G$ the center-of-mass kinetic energy operator. $V_{ij}^{(2)}$ and $V_{ijk}^{(3)}$ are the effective two-body and three-body nuclear interaction energy, respectively. $V_{ij}^C$ represents the Coulomb interaction energy between protons.

We perform the GCM calculations for the $5\alpha$ cluster system. The key size variables can be treated as generator coordinates and we take mesh points for $\{\beta_1^{(i)}, \beta_2^{(i)}\}$ from 0.5 fm to 12.5 fm with the step 1.0 fm in the GCM calculations.

$$\Psi_{\text{gcm}}^{0_\lambda^+}(^{20}\text{Ne}) = \sum_{i=1}^{169} C_i^\lambda \, \Psi(\beta_1^{(i)}, \beta_2^{(i)}). \tag{7}$$

Superposition of total 169 spherical two-$\beta$ THSR wave functions and solving the Hill-Wheeler equation, their diagonalization yield −126.9 MeV and −156.4 MeV energies for the ground state of $^{16}$O and $^{20}$Ne respectively, which agree with the corresponding experimental values of −127.6 MeV and −160.6 MeV. Indeed, these energies can be further improved if more $\alpha$ correlations and deformations are taken into account.

## Treatment of resonance states

Above the $5\alpha$ threshold, the continuum states can hardly be avoided after superposing many different configurations. To identify the required resonance states, the radius-constraint method[48] in our microscopic cluster model is applied. We diagonalize the squared radius operator and obtain the corresponding eigenstates and eigenvalues. The radius eigenfunctions whose eigenvalues are smaller than the cutoff parameter $R_{\text{cut}}$ can be our basis wave functions in GCM calculations. This method is essentially similar with the finite-volume method[49] in other models. Moreover, the stabilization method in the theory of resonances has the consequence that, except for special broad cases, the obtained eigen energies of resonance states hardly change due to the slow increase of the $R_{\text{cut}}$, which is the bounded volume or barrier, while the continuum states change dramatically. Therefore we can deal with the continuum states and approximate the resonant states in our calculations. In Supplementary Figure 1 we show the dependence of $R_{\text{cut}}$ for all obtained $0^+$ states in $^{20}$Ne.

## Reduced width amplitude and $S^2$ factor

Based on the obtained GCM wave functions, the reduced width amplitudes can be calculated,

$$y(a) = \sqrt{\frac{20!}{4!16!}} \left\langle \left[ [\Psi_{\text{gcm}}^{0_s^+}(^{16}\text{O})\varphi_5(\alpha)]_{0^+} Y_{00}(\hat{\xi}_4) \right]_{0^+} \frac{\delta(\xi_4 - a)}{\xi_4^2} \middle| \Psi_{\text{gcm}}^{0_\lambda^+}(^{20}\text{Ne}) \right\rangle, \tag{8}$$

where $\xi_4$ is the dynamic coordinate of relative motion between the center-of-mass coordinates of $\alpha$ cluster and $^{16}$O cluster. $\Psi_{\text{gcm}}^{0_s^+}(^{16}\text{O})$ and $\Psi_{\text{gcm}}^{0_s^+}(^{20}\text{Ne})$ are the obtained $s_{th}$ and $\lambda_{th}$ eigen wave functions for $^{16}$O ($^{12}$C+ $\alpha$) and $^{20}$Ne ($^{16}$O + $\alpha$), respectively. The corresponding

spectroscopic $S^2$ factor of $^{16}$O + $\alpha$ component is defined as follows,

$$S^2 = \int_0^{+\infty} dr \, [ry(r)]^2. \tag{9}$$

From the RWA and spectroscopic $S^2$ factor, the partial $\alpha$ decay width can be calculated based on $R$ matrix theory. In addition, much structure information of $^{20}$Ne can be obtained from the RWA, which characterizes the relative motion of $\alpha$ and $^{16}$O clusters.

## States above $5\alpha$ threshold

We focus only on $5\alpha$ cluster states above threshold, where the obtained $0_{15-19}^+$ states taking $R_{\text{cut}} = 10$ fm are discussed here. It can be seen that the $0_{15}^+$ state has a very large component of the $^{16}$O $(0_2^+) + \alpha$ configuration and shows a strong oscillatory behavior, especially in the outer region ($a > 8$ fm) from the RWA (See Supplementary Figure 3). This is the typical behavioral character of the continuum state. As for the $0_{16}^+$ state, it is above the $5\alpha$ threshold but has a non-negligible component of the $^{16}$O $(0_1^+) + \alpha$ structure and could also contain some continuum states (see Supplementary Figure 2). Moreover, the Supplementary Figure 4 shows that the largest peak in the RWA of $^{16}$O $(0_6^+) + \alpha$ is in the $0_{18}^+$ state as far as 19 fm and this is clearly the unphysical state. Thus, to determine the possible condensate state, we only need to consider the $0_{17}^+$ and $0_{19}^+$ states above the $5\alpha$ threshold, which are denoted as $0_{\text{I}}^+$ and $0_{\text{II}}^+$ states, respectively. Strikingly, the $0_{17}^+$ ($0_{\text{I}}^+$) state ($E_x \approx 22$ MeV) is relatively stable and shows little dependence on $R_{\text{cut}}$ in our calculations. For example, taking $R_{\text{cut}} = 8$ fm and 10 fm, the obtained eigen energies are almost identical as shown in Supplementary Figure 1, and the corresponding overlap values with their GCM wave functions are as high as 0.95.

## Data availability

All data relevant to this study are shown in the paper and its Supplementary file, and more details are available from the corresponding authors.

## Code availability

Inquiries about the code in this work will be responded to by the corresponding authors.

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

## Acknowledgements

We acknowledge discussions with Z. Ren, C. Xu, M. Kimura, M. Lyu, Y. Ye, Z. Yang, Q. Zhao, X. Cao, T. Myo, and H. Toki. This work was supported in part by the National Natural Science Foundation of China under contract Nos. 12175042, 11890710, 11890714, 12047514, and 12147101, Guangdong Major Project of Basic and Applied Basic Research No. 2020B0301030008, and China National Key R&D Program No. 2022YFA1602402. This work was partially supported by the 111 Project. This work was supported in part by Shanghai "Science and Technology Innovation Action Plan" Project No. 21ZR1409500. This work was supported by RCNP Collaboration Research Network (COREnet). This work was supported by MEXT/JSPS KAKENHI Grant Number JP21H00127. Numerical computation in this work was partly carried out at the Yukawa Institute Computer Facility.

## Author contributions

B.Z. and Y.F. proposed this project and performed the calculations. B.Z. prepared the manuscript with help from Y.F. and Y.G.M. All authors, including B.Z., Y.F., H.H., Y.G.M., G.R., P.S., A.T., and T.Y., contributed to the discussion of results and were involved in revising the manuscript.

## Competing interests

The authors declare no competing interests.
