## [Peer Review File · Nature Communications]

The 5α condensate state in ^{20}NeREVIEWER COMMENTS

Reviewer #1 (Remarks to the Author):

This paper presents a novel microscopic 5-body calculation for ^{20}Ne using a new wavefunction to investigate a 5α nuclear structure in ^{20}Ne . Using these microscopic calculations, this work details the possible existence of a 5α condensate state ^{20}Ne at around 3 MeV excitation energy, and shows possible evidence in available experimental data to support the existence of such a state. The possible existence of such a 5α -condensate state is of great importance to the nuclear physics community to improve the understanding of fundamental nuclear interactions in finite nuclei, and the α -clustering phenomenon. Possible experimental signatures provided by this work, which could be observed to prove the existence of such a condensate state, will definitely influence the nuclear physics community to push the boundaries on scientific research in this field. As the authors have correctly pointed out, obtaining experimental proof of a 4α condensate state in ^{16}O is still very challenging, and a possible 5α -condensate state would hopefully be less so. This manuscript is well-written, and the claims made towards the possible existence of a 5α condensate state in ^{20}Ne is sound. As such, I view this work to have a high importance in understanding the $N\alpha$ clustering structure and in identifying α -condensate states in finite nuclei. It definitely warrants publication in Nature Communications. Before publication, the authors should address the comments below.

In line 81 and onwards until about halfway of the main bulk of the manuscript, the authors have written the manuscript in such a way that makes it sound like only 2 of the 19 0^+ states obtained from the microscopic calculations are above the 5α threshold, when there are indeed 5. I believe a bit more clarification on this is necessary in the beginning (and also perhaps in fig. 2), whereas the authors describe this only at the very end in the Methods section in "States above 5α threshold".

The approach of using the spectroscopic factors for different channels in the two main 0^+ states of ^{20}Ne above the 5α threshold is a good approach to validate the possible structure of a 5α condensate state. Looking at the Supplementary figure 3, the $0^+(15)$ shows a peak in the RWA within a reasonable distance of about 10 fm for the $^{16}\text{O}(0^+(2))+\alpha$ channel. Similarly in the Supplementary figure 2, the same is valid for $0^+(16)$ for the $^{16}\text{O}(0^+(1))+\alpha$ channel. For conciseness, I wonder what these would relate to in the spectroscopic factor plot of Figure 2. Maybe the spectroscopic factors of the different channels can be shown in figure 2 for the 0^+ states of 15 and 16 as a comparison, in addition to 17 and 19 to further substantiate the $^{16}\text{O}(0^+(2))+\alpha$ structure claim for the 17 and 19 0^+ states.

The authors have showed that the theoretical approach of a large overlap between the proposed wavefunction with the single one-beta THSR wavefunctions can be used to identify an α -condensate state. The authors have stated in line 143 that "...most 0^+ states have a quite small overlap.. due to the non- 5α clustering structure". But the authors fail to comment on the relatively non-zero overlap seen in

the 11th and 13th 0^+ states in Figure 5, and how they could possibly contribute (or be connected) to the $0^+(17)$ candidate for the 5α -condensate state.

The heading of the 4th column of table 1: better to be written the other way around as $0^+(II) \rightarrow 0^+(I)$.

Reviewer #2 (Remarks to the Author):

The paper '5a condensate..' discusses the possibility of a boson condensate in alpha clustered nuclei and in particular ^{20}Ne . Predictions are made for states above the Na decay threshold and some comparison to data is discussed. The paper is well written and the results are reasonable. The paper is suitable for publication after some minor misprints are corrected.

We appreciate the comments, suggestions, and recommendations from the reviewers regarding our manuscript. In the revised manuscript, we have addressed all of the concerns raised by the reviewers, which we believe has much enhanced the quality of our study. Detailed responses and revisions corresponding to each comment are provided below.

#Reviewer 1

Comment1:

In line 81 and onwards until about halfway of the main bulk of the manuscript, the authors have written the manuscript in such a way that makes it sound like only 2 of the 19 0^+ states obtained from the microscopic calculations are above the 5α threshold, when there are indeed 5. I believe a bit more clarification on this is necessary in the beginning (and also perhaps in fig. 2), whereas the authors describe this only at the very end in the Methods section in "States above 5α threshold".

Response1:

Yes, we agree with this point and thanks for this suggestion. To enhance the coherence of the manuscript as suggested by the reviewer, we made further clarification by adding one paragraph from line 79.

“We perform full-microscopic calculations for the 5α structure and obtained 19 states of $J^\pi = 0^+$ in ^{20}Ne . Among the obtained 19 states, the most significant ones are those above the 5α threshold, as they are potential candidates for the 5α condensate state. Our current calculations yield five 0^+ states (0_{15-19}^+ states) above the threshold. However, two of these five, 0_{17}^+ and 0_{19}^+ eigenstates, are only reasonably considered as candidates for 5α cluster states. The remaining three 0^+ states are considered to be unphysical ones with the continuum states contaminated, based on the radius constraint method and analysis of their reduced width amplitudes in different channels. For further details, please refer to the method section. In the next discussions, we only need to focus on the 0_{17}^+ and 0_{19}^+ states, which are denoted as 0_{I}^+ and 0_{II}^+ states as shown in Fig.2.”

***Comment2:** The approach of using the spectroscopic factors for different channels in the two main 0^+ states of ^{20}Ne above the 5α threshold is a good approach to validate the possible structure of a 5α condensate state. Looking at the Supplementary figure 3, the $0^+(15)$ shows a peak in the RWA within a reasonable distance of about 10 fm for the $^{16}\text{O}(0^+(2))+\alpha$ channel. Similarly in the Supplementary figure 2, the same is valid for $0^+(16)$ for the $^{16}\text{O}(0^+(1))+\alpha$ channel. For conciseness, I wonder what these would relate to in the spectroscopic factor plot of Figure 2. Maybe the spectroscopic factors of the different channels can be shown in figure 2 for the 0^+ states of 15 and 16 as a comparison, in addition to 17 and 19 to further substantiate the $^{16}\text{O}(0^+(2))+\alpha$ structure claim for the 17 and 19 0^+ states.*

Response2: Thank you for pointing this out and for your suggestions. Yes, the obtained 0_{15}^+ state has a very large component of $^{16}\text{O}(0_2^+)+\alpha$ structure with a strong oscillatory character from the calculated RWA. Moreover, this state is located at as high as about 7 MeV above the $^{16}\text{O}(0_2^+)+\alpha$ threshold and higher than the corresponding Coulomb barrier. Clearly, this state is a continuum state of the two-body $^{16}\text{O}(0_2^+)+\alpha$ state. Similarly, the 0_{16}^+ state, primarily composed of $^{16}\text{O}(\text{g.s.})+\alpha$ configuration, is located as high as about 19 MeV above the $^{16}\text{O}(\text{g.s.})+\alpha$ threshold, a position that is far above the Coulomb barrier. Regarding the 0_{18}^+ state, its main configuration is the $^{16}\text{O}(0_6^+)+\alpha$ structure from the S^2 factor shown in Fig. A. Nevertheless, the outermost peak of the

corresponding RWA for this channel, which has the largest amplitude, is located around 19 fm (see supplementary Fig.4). These facts indicate the 0_{16}^+ and 0_{18}^+ states are contaminated by the continuum states. Therefore, we did not show any energies, S^2 factors quantities for these three unphysical states in Fig.2 and Fig.3. Our focus in this work is on the possible 5α cluster states, and these three states are clearly not potential 5α gas-like states.

Figure A: Based on the obtained wave functions of the ground state and five 0^+ states of ^{20}Ne above the 5α threshold, the spectroscopic S^2 factor for three channels of $^{16}\text{O} + \alpha$ are calculated. (Note that the 0_{15}^+ , 0_{16}^+ , and 0_{18}^+ states are not the physical states.)

Based on the reviewer's suggestion, to make a comparison, we calculated the S^2 factors for the other three 0^+ states as well. Figure A shows the calculated S^2 factors for the ground state and the obtained five 0^+ states above the 5α threshold in three channels. It shows the different components of the six 0^+ states within three channels, particularly, the very large $^{16}\text{O} (0_2^+) + \alpha$ configuration for the 0_{15}^+ state is shown. Of course, all these information have been included in the RWA of supplementary Figs. 2-4 and actually their RWA can provide us more details of behaviors of the wave functions and their components. Most importantly, the 0_{15}^+ , 0_{16}^+ , and 0_{18}^+ states are not the physical states. Therefore, we only discuss the two 0^+ states in the main text while show all the RWA for the five 0^+ states as the supplementary information.

Comment3: *The authors have showed that the theoretical approach of a large overlap between the proposed wavefunction with the single one-beta THSR wavefunctions can be used to identify an α -condensate state. The authors have stated in line 143 that "...most 0^+ states have a quite small overlap.. due to the non- 5α clustering structure". But the authors fail to comment on the relatively non-zero overlap seen in the 11th and 13th 0^+ states in Figure 5, and how they could possibly contribute (or be connected) to the $0^+(17)$ candidate for the 5α -condensate state.*

Response3: Thank you for this comment. Yes, from the Fig.5, below the 5α threshold, besides the ground state and 0_3^+ state, the 0_{11}^+ and 0_{13}^+ states also show some non-negligible overlap (max. around 0.4) with the single one-beta THSR wave function as the reviewer mentioned. We haven't mentioned this point in our manuscript. In fact, all these states can be considered as the intermediate states to evolve into the final gas-like condensate states. We revised the corresponding text.

Comment4: *The heading of the 4th column of table 1: better to be written the other way around as $0+(II) \rightarrow 0+(I)$.*

Response4: Yes, that would be better. We have changed it to $0+(II) \rightarrow 0+(I)$.

#Reviewer 2

Comment : *The paper '5a condensate..' discusses the possibility of a boson condensate in alpha clustered nuclei and in particular ^{20}Ne . Predictions are made for states above the Na decay threshold and some comparison to data is discussed. The paper is well written and the results are reasonable. The paper is suitable for publication after some minor misprints are corrected.*

Response: Thanks for your recommendations. We have carefully read the manuscript and made some minor revisions. Please refer to the manuscript text file, where changes are highlighted in red.

REVIEWERS' COMMENTS

Reviewer #1 (Remarks to the Author):

The responses to the previous comments and the revisions made to the manuscript by the authors are robust. The revisions strengthen the submission further. I would agree that this manuscript can now be published in Nature Communications, as it addresses an important area in nuclear physics.